# Occurrence of Mycosporine-like Amino Acids (MAAs) from the Bloom-Forming Cyanobacteria *Aphanizomenon* Strains

**DOI:** 10.3390/molecules27051734

**Published:** 2022-03-07

**Authors:** Hang Zhang, Yongguang Jiang, Chi Zhou, Youxin Chen, Gongliang Yu, Liping Zheng, Honglin Guan, Renhui Li

**Affiliations:** 1Hubei Water Resources Research Institute, Hubei Water Resources and Hydropower Science and Technology Information Center, Wuhan 430070, China; hungryzhang@163.com; 2Department of Biological Sciences and Technology, School of Environmental Studies, China University of Geosciences, Wuhan 430074, China; jiangyg@cug.edu.cn; 3Hubei Water Resources Research Institute, Hubei Water Resources and Hydropower Science and Technology Promotion Center, Wuhan 430070, China; zhouc0222@126.com; 4Key Laboratory of Algal Biology, Institute of Hydrobiology, Chinese Academy of Sciences, Wuhan 430072, China; chenyouxin@ihb.ac.cn (Y.C.); yugl@ihb.ac.cn (G.Y.); 5BGI Genomics Co., Ltd., Wuhan 430072, China; zhengliping@genomics.cn; 6College of Life and Environmental Sciences, Wenzhou University, Wenzhou 325000, China

**Keywords:** *Aphanizomenon* strains, bloom-forming cyanobacteria, morphological and phylogenetic analysis, MAAs, identification and quantification

## Abstract

Mycosporine-like amino acids (MAAs) are widespread in various microbes and protect them against harsh environments. Here, four different *Aphanizomenon* species were isolated from severely eutrophic waterbodies, Lake Dianchi and the Guanqiao fishpond. Morphological characters and molecular phylogenetic analysis verified that the CHAB5919, 5921, and 5926 strains belonged to the *Aphanizomenon flos-aquae* clade while Guanqiao01 belonged to the *Aphanizomenon* gracile clade. Full wavelength scanning proved that there was obvious maximal absorption at 334 nm through purified methanol extraction, and these substances were further analyzed by HPLC and UPLC-MS-MS. The results showed that two kinds of MAAs were discovered in the cultured *Aphanizomenon* strains. One molecular weight was 333.28 and the other was 347.25, and the daughter fragment patterns were in accordance with the previously articles reported shinorine and porphyra-334 ion characters. The concentration of the MAAs was calibrated from semi-prepared MAAs standards from dry cells of *Microcystis aeruginosa* PCC7806 algal powder, and the purity of shinorine and porphyra-334 were 90.2% and 85.4%, respectively. The average concentrations of shinorine and porphyra-334 were 0.307–0.385 µg/mg and 0.111–0.136 µg/mg in *Aphanizomenon flos-aquae* species, respectively. And there was only one kind of MAAs (shinorine) in *Aphanizomenon gracile* species.,with a content of 0.003–0.049 µg/mg dry weight among all *Aphanizomenon gracile* strains. The shinorine concentration in *Aphanizomenon flos-aquae* was higher than that in *Aphanizomenon gracile* strains. The total MAAs production can be ranked as *Aphanizomenon flos-aquae* > *Aphanizomenon gracile*.

## 1. Introduction

Solar ultraviolet (UV) radiation is harmful to organisms as it causes damage to biological macromolecules [1]. However, microbes have developed a combination of strategies to deal with UV exposure such as the biosynthesis of sunscreens for UV photoprotection [2,3,4,5,6,7]. Mycosporine-like amino acids (MAAs), are water-soluble, UV-absorbing substances composed of a cyclohexenone or a cyclohexenimine group conjugated with amino acids or amino alcohols [8,9]. MAAs can efficiently dissipate the absorbed energy of UV radiation as heat without the generation of reactive oxygen species (ROS) [10]. The strong UV-absorption activity and the free-radical scavenging ability of MAAs make them the most promising natural products in biotechnological and biomedical industries [7,11,12].

Although MAAs were firstly discovered in fungi and associated with UV-induced sporulation and named accordingly [13], more MAAs have been found in various organisms, including bacteria, micro- and macro-algae, and terrestrial lichens from tropical coral reefs to polar glaciers [14,15,16,17]. Other aquatic animals such as zooplanktons, marine invertebrates, and fish also accumulate MAAs via the algal route [18]. Thus, MAAs protect not only the producers but also the consumers from UV radiation through the food chain [19]. So far, more than 74 MAAs with different substituents have been discovered [8,17,20].

Bloom-forming cyanobacteria have been well characterized as possessing gas vesicles facilitating their floating and bloom- forming on water surfaces [21,22,23]. However, such a competitive strategy increases their exposure to UV radiation, so it is necessary for these bloom-forming cyanobacteria to develop their ability to produce MAAs for self-protection. Indeed, both natural bloom samples dominated by *Microcystis* strains have been shown to contain shinorine and porphyra-334 [24,25,26] and the MAA concentration was significantly correlated with the occurrence of *Microcystis* but not with other algal groups [27]. Moreover, the extracts from *Aphanizomenon* harvested from the blooms in the Upper Klamath Lake, US, were also shown to contain MAAs and showed UV-A protection [28]. *Aphanizomenon* was a dominant species along with *Microcystis* spp. in the cyanobacterial blooms of Lake Dianchi Lake, China, a high plateau lake with a high intensity of sunshine and UV radiation. Moreover, early spring *Aphanizomenon* blooms have occurred annually in Lake Dianchi during the last decade [29,30]. Therefore, the characterization of the UV-absorbing substances in the *Aphanizomenon* strains will help us to better understand the competitive strategy and bloom formation mechanism of this cyanobacterium.

In this study, five bloom-forming *Aphanizomenon* strains isolated from Lake Dianchi and the Guanqiao fishpond were isolated and verified according to their morphological character and phylogenetic analysis, and then the MAAs in these strains were examined and purified. The results showed that all the tested *Aphanizomenon* strains can synthesize one or two kinds of MAAs. All the MAAs were quantified and collected as standards for further studies.

## 2. Materials and Methods

### 2.1. Aphanizomenon Strains and Culture Conditions

Five *Aphanizonmenon* (*Aph.*) strains were examined in this study, including strains CHAB5919, CHAB5921, CHAB5926, FACHB1039, and Guanqiao01. The former four strains were isolated from Lake Dianchi in Kunming city, and Guanqiao01 was collected from the Guaoqiao fishpond in Wuhan. Pure cultures of these strains were obtained by the Pasteur pipette method [31] and preserved in the Culture Collection of Harmful Algae Biology, Institute of Hydrobiology, the Chinese Academy of Sciences. All strains were maintained in CT medium [32] at 25 °C with a constant white light intensity of 30 µmol photons m^−2^·s^−1^ under a 12 h light:12 h dark cycle [33]. For MAA analysis, the strains were inoculated in a 10 L medium in a serum bottle. *Aphanizomenon* cells were harvested by centrifugation at 5000 rpm/min for 20 min when the OD_750_ nm of the culture was 1.0. The cell pellets were frozen at −80 °C for at least 6 h before lyophilization over light.

### 2.2. DNA Extraction and PCR Reaction

The genomic DNA of all the *Aphanizomenon* strains was extracted according to the previous methods [34] with minor modifications. Briefly, collected fresh algal cell pellets were resuspended in the lysis buffer (100 mmol/L Tris-HCl, 100 mmol/L NaCl, 1 mmol EDTA, pH 9.0) containing lysozyme (10 mg/mL) and incubated at 37 °C for 30 min. The proteinase K (20 mg/mL) and 10% (*v*:*v*) SDS were added to the mixtures and incubated at 55°C for two hours. The sample was further extracted with a mixture of phenol-chloroform-isoamylol (25:24:1 *v*:*v*:*v*), and centrifuged at 12,000 rpm for 10 min. The supernatant was taken, and the DNA was precipitated by sodium acetate (10 M) and ethanol. After washing with 70% ethanol solution, the DNA samples were dissolved in bidistilled water and stored at −20 °C.

The primer pair of F1 (5′-TTGATCCTGGCTCAGGATGA-3′) [35] and 1480 (5′-AGTCCTACCTTAGGCATCCCCCTCC-3′) [36] was used to amplify the 16S rRNA gene sequences of the *Aphanizomenon* strains. All PCR reactions were performed in a volume of 20 µL mixture containing 2 µL 10× PCR buffer (Takara, Japan), 500 µmol dNTPs Mix, 200 pmol of each primer, 100 ng genome DNA, and 1U Pfu UltraTag DNA polymerase (Takara, Japan) with pure water added to reach the final volume. PCRs were run in a Biorad Thermal Cycler (Biorad, Hercules, CA, USA) with the following cycle: 95 °C for 5 min, 34 cycles of 95 °C for 30 s, Tm for 30 s, 72 °C for 45 s, final elongation at 72 °C for 5 min, and cooling to 4 °C. PCR products were detected by electrophoresis. Target fragments were purified with the Bioflux Gel DNA extraction kit according to the manufacturer’s protocol and sequenced with an ABI 3730 automated sequencer (Perkin-Elmer, Waltham, MA, USA). All sequences were submitted to the Genbank database, and the accession numbers are shown in Table 1.

### 2.3. Sequence Alignment and Phylogenetic Analysis

The genetic evolution relationship of *Aphanizomenon* was analyzed using their 16S rRNA sequences. Five new 16S rRNA gene sequences in this study and seventeen previous 16S rRNAs of *Aphanizomenon*-related species obtained from Genbank were used to construct the phylogenetic tree. *Aphanizomenon* 16S rRNA sequences from this study and the GenBank database were aligned with the software Bioedit Sequence Alignment Editor Version 7.1.3.0 (Abbott laboratories, Lake Bluff, IL, USA) [37], and manually corrected.

Phylogenetic analysis was conducted using MEGA version 6.0 (Research center for genomics and bioinformatics, Tokyo Metropolitan University, Tokyo, Japan) [38] and the neighbor-joining (NJ) algorithm. Kimura [39] was used to estimate the phylogenetic distances of each sequence and the robustness of each node on the NJ tree was evaluated by the bootstrap method with 1000 replications. *Cylindrospermopsis raciborski* CHAB3438 with a filamentous and heterocytous character was set as the outgroup for the phylogenetic analysis of the 16S rRNA.

### 2.4. MAA Extraction and HPLC Analysis

One gram of lyophilized cells was placed in a 10 mL centrifuge tube, and 5 mL of HPLC-grade methanol was added. The extraction of the MAAs was performed by ultrasonication for 5 min in an ice bath before the methanol solution was placed in the dark at 4 °C for 12 h. After centrifugation at 12,000 rpm for 10 min, the green supernatant was taken and desiccated by rotary evaporation at 45 °C for 2 h. The dried residues were dissolved in bidistilled water, and a half volume of chloroform was added to remove the pigments and polysaccharides. The mixture was centrifuged at 12,000 rpm for 15 min, and the supernatants were filtered through a 3KD ultra-filtration tube (Millipore, Burlington, MA, USA) at 5000 rpm for 20 min. Before HPLC analysis, the filtrates were scanned using a UV-2550 spectrophotometer (Shimadzu, Japan) at an absorption spectrum from 200 nm to 750 nm.

MAA detection was conducted on a HPLC (SSI series 1550, Pennsylvania State University, USA) equipped with a UV-VIS detector. A Synergi 4µ Fusion-RP 80A column (250 × 4.6 mm, Phenomenex) conjugated with a guard pre-column containing the same material was used for separation. The detection wavelength was set at 330 nm. Isocratic elution was performed by a mobile phase of 25% (*v*:*v*) methanol and 0.1% (*v*:*v*) acetic acids at a flow rate of 0.7 mL/min. A total of 20 µL of the sample was injected and the column heater temperature was maintained at 30 °C.

The MAAs were identified by comparison with published retention times and co-chromatography with purified standards extracted from *Microcystis aeruginosa* PCC7806. The peak purity was checked by the whole spectrum absorption over the entire wavelength. The specific MAA concentration of each sample was calculated from the HPLC peak area and the extension of Lanbert Beer’s Law A = ε·b·c, using published molar extinction coefficients [40,41,42,43,44], for shinorine (ε = 44,700 M^−1^·cm^−1^), and porphyra-334 (ε = 42,300 M^−1^·cm^−1^). The MAA concentrations are expressed as µg/mg dry weight alga powder.

### 2.5. UPLC-MS-MS Analysis

MAAs were also identified using Waters Acquity UPLC-H-class-Xevo TQ MS linked with Tandem Quadrupole (Triple Quadrupole) mass spectrometry (Waters, Wexford, Ireland). The separation was performed with Waters Acquity UPLC BEH C18 (2.1 × 50 mm, 1.7 µM particles) whose temperature was maintained at 40 °C. A total of 10 µL of the sample was injected into the auto-sampler, which was kept at room temperature. The flow rate was 200 µL/min with the mobile phase methanol containing 0.1% (*v*:*v*) formic acid under isocratic conditions. All samples were detected in the multiple reaction monitoring mode (MRM) and ESI-positive mode. The ESI source temperature and desolvation temperature were set as 150°C and 450 °C, respectively. The capillary voltage and the extraction voltage were set at 3.0 kV and 5 V, respectively. Argon was used as collision gas at the flow rate of 0.2 mL/min, and nitrogen gas was used as the cone gas and desolvation gas. The flow rate of the cone gas and desolvation gas rate were 20 L/h and 650 L/h, respectively. The system operation and data acquisition were controlled using the software of MassLynx V4.1 (Waters, Ireland).

### 2.6. Collection of MAA Standards

*Microcystis aeruginosa* PCC 7806 can produce two kinds of MAAs: shinorine and porphya-334, with the maximal absorption at 334 nm [25]. Therefore, the dried cell powder of *M. aeruginosa* PCC7806 was used for the preparation of MAA standards. The semi-preparation column Synergi 4µ Fusion-RP 80A column (250 × 10 mm, Phenomenex) was used to separate the two kinds of MAAs. A total of 500 µL of the aqueous sample was injected and the flow rate was set at 5.0 mL/min with the mobile phase of 25% methanol containing 0.1% acetic acids under isocratic conditions. The detection wavelength was set as 330 nm. The MAA eluates were collected and lyophilized according to their retention time. The pellets were dissolved by 25% methanol. All the aqueous solutions were transferred to a Strata-X 33 µ high-performance polymeric solid phase extraction column (200 mg/6 mL, Phenomenex). Solid phase extraction was conducted according to the protocols set by the manufacturer. The last eluate containing concentrated MAAs was collected and used as a standard.

## 3. Results

### 3.1. Morphological and Phylogenetic Classification of Aphanizomenon Strains

All the strains were examined under a microscope. As shown in Figure 1, the three strains from Lake Dianchi, CHAB 5919, CHAB 5921, and CHAB 5926 were shown to have typical fascicle-like assemblages with numerous trichomes, and terminal cells were developed as hyaline, elongated as cylindrical, the length was 2–3 times longer than the width of the vegetative cell; therefore, these strains were morphologically identified as *Aph. flos-aquae*. The strain Guanqiao01 was shown as single filaments with blunt round terminal cells, and the cells’ length was nearly the same as the width; thus, it was identified as *Aph. gracile* [33]. (Figure 1).

A phylogenetic tree was conducted according to the 16S rRNA sequences of the strains, and all the *Aphanizomenon* species were divided into three independent clades: the *Aph. flos-aquae*, *Aph. gracile*, and *Cus. issatschenkoi* clade, respectively. The strains CHAB5919, CHAB5921, and CHAB5926 were classified into the *Aph. flos-aquae* clade as their gene sequence similarity was up to 95–99% when compared with other published *Aph. flos-aquae* strains, and the *Aph. gracile* clade gene sequence similarity was about 96%. The strains FACHB1039 and Guanqiao01 were clustered to the *Aph. gracile* clade (Figure 2). These results demonstrate that the morphological identification of these strains was reliable.

### 3.2. Discovery of a UV Absorption Substance in the Aphanizomenon Strains

In order to determine the reliability of our MAA extraction and HPLC analysis methods, the extracts from *Aph. flos-aquae* CHAB5921 and *M. aeruginosa* PCC7806 were compared. The full wavelength absorption spectra of these two strains were similar, with the same maximal absorption at 334 nm, which was within the ultraviolet scale, but *M. aeruginosa* PCC7806 had a higher peak absorbance than *Aph. flos-aquae* CHAB5921 with the same dry weight. The results prove that there was a UV absorption substance in the *Aphanizomenon* strains (Figure 3a).

### 3.3. HPLC Analysis of the UV Absorption Substance

The HPLC analysis showed that *Aph. flos-aquae* CHAB5921 and *M. aeruginosa* PCC7806 had two of the same peak absorptions at 334 nm with a retention time at 9.7 min and 11.3 min, respectively. (Figure 4). However, the substance concentrations between these two strains were different: the eluate I contents in *M. aeruginosa* PCC7806 was about 25 times more than *Aph. flos-aquae* CHAB5921, but the content eluate II in *M. aeruginosa* PCC7806 was 3 times less than *Aph. flos-aquae* CHAB5921. Both the substance concentrations showed an extraordinary distinction.

### 3.4. Identification of MAAs in the Aphanizomenon Strains

The mixture of purified compounds from eluates corresponding to the above two peaks from *Aph. flos-aquae* CHAB5921 and *M. aeruginosa* PCC7806 also showed the same absorption spectrum (Figure 3b). Analysis by UPLC-MS-MS revealed that the extracts from *Aph. flos-aquae* CHAB5921 had two dominant positive ions with *m*/*z* 333.28 and 347.25 (Figure 5A) These two ions were in accordance with the molecular weights of the two MAAs, shinorine and porphyra-334, respectively (Table 2). Further analysis showed that the daughter fragments of *m*/*z* = 333 (Figure 5B.) and *m*/*z* = 347 (Figure 5C) were also the same as the previously reported fragments of shinorine and porphyra-334 [19,45].

### 3.5. Quantification of MAAs in Different Aphanizomenon Strains

The semi-preparation column in HPLC was used to purify the different fractions of MAA in *M. aeruginosa* PCC7806. 18.7 mg of shinorine and 2 mg of porphyra-334 were finally obtained and used as standards. The purities of shinorine and porphr-334 were 90.2% and 85.4%, respectively. The concentrations of the MAAs of different *Aphanizomenon* strains were detected and are shown in Table 3. All the three *Aph. flos-aquae* strains could synthesize two kinds of MAAs, with the average shinorine concentration being 0.307–0.385 µg/mg, and the average porphyra-334 concentration being 0.111–0.136 µg/mg, with the highest total concentration being 0.443–0.497 µg/mg dry weight. The *Aph. gracile* strains only synthesized shinorine with a concentration of 0.003–0.049 µg/mg; their MAA productivity was lower than the *Aph. flos-aquae* strains.

## 4. Discussion

The classification of *Aphanizomenon* species is not an easy task. Before the molecular–biological methods were brought into the classification system, an alga was named according to its morphological character, and the *Aphanizomenon* strains’ main feature changed when the culture condition varied: the typical fascicle-like assemblages in *Aphanizomenon flos-aquae* strains may become a single filament, making it difficult to name the algal species. The *Aphanizomenon* PMC9706, PMC9501, and NIES81 strains were once named *Aph. flos-aquae* according to their morphological character, but afterwards molecular–biological gene sequences proved that these strains should be classified into the *Aph.gacile* clade. For the purpose of respecting predecessors’ work, articles afterwards did not change their original name, but the molecular–biological technique verified that *Aph. flos-aquae* PMC9706, PMC9501, and NIES81 were actually divided into the *Aph.gracle* claden [33]. The slight mixture of these phenomena did not affect the last conclusion.

Previous studies have reported the production of MAAs in various cyanobacteria [5,8,14,16]. These studies mainly focused on the protective role of the MAAs against UV radiation [46,47,48,49]. The cyanobacteria have been shown to be efficient producers of MAAs [50], but the occurrence of MAAs in water-bloom cyanobacteria has rarely been reported [24,27].

Liu et al. discovered two kinds of MAAs (shinorine and porphyra-334) from both a culture and a natural sample of *M. aeruginosa* from Lake Taihu with the detection of the absorption spectrum and HPLC analysis, and these substances were supposed to be an important factor that promoted *M. aeruginosa* as a dominant surface bloom-forming species, but the article did not recover the concentration of the main MAAs, and the two different MAAs were difficult to elute from those conditions [24]. Based on a photo-optical model, MAAs were identified by comparison with the published retention time and by co-chromatography with a purified shinorine standard obtained from laboratory donation and marine algal extracts. It was estimated that the highest MAA concentration could be 2.5% of the dry weight with the ability to confer 40% of internal screening to a single layer of microcystis cells; thus, the *M. aeruginosa* can form a colony with several layers by internal self-shading and can use a combination of MAA strategies of, using carotenoid and D-galacturonic acid in a slim layer to cope with the high intensity of UV radiation [27]. Moreover, the *Dolichospermum floa-aquae* CHAB1629 from Lake Taihu has also been proven to synthesize MAAs in enlarging cultures [51].

However, the exact role of MAAs in different *Microsystis* varies. Paerl et al. did not find any MAAs in a summer bloom of *M. aeruginosa*, but the cellular carotenoid contents increased via the presence of surface photosynthesis, so maybe the extraction reagent was not that efficient in the extraction of these substances and the other pigments in the cells may also have masked the absorption of MAAs [27,52]. A bloom-forming cyanobacterium, *M. aeruginosa* 854, was shown to cope with enhanced UV-B through increasing the synthesis of carotenoids and degrading polysaccharides against oxidative stress to avoid further damage to DNA. They used the same extraction protocols, but the MAAs did not occur in this bloom-forming species [53]. The MAA of shinorine was also discovered in a pure culture sample of *M. aeruginosa* PCC7806, but the experiment suggested that the MAAs in these strains play a negligible role in the protection against UV radiation but maybe have a strain-specific trait involved in the formation of an extracellular matrix and cell–cell interaction [25].

Porphyra-334 was the first to be discovered in commercial *Aph. flos-aquae* capsules and showed UV-A protection activities [28]. A high-resolution NMR technique was used to identify the MAAs from *Aph. flos-aquae* in Kalamath Lake dry powder [28,54]. In the above-mentioned articles, the discovery of MAAs was from the outside samples; there may be many other uncertainties that exist that interfered with the judgement on whether the MAAs came from the single cyanobacteria or other mixture microalgae, so a pure culture of the *Aphanizomenon* strain was needed to evaluate their MAA production possibility.

In this study, four *Aphanizomenon* strains isolated from a severe eutrophic waterbody belonged to the *Aph. flos-aquae* clade and *Aph. gracle* clade and were verified by their morphological character and phylogenetic molecular analysis. Pure *Aphanizomenon* strains were obtained and enlarged cultivation. The MAA extraction methods were based on the average protocols with little modification [55]. Based on the conjugated techniques of the absorption spectrum, HPLC elution, and UPLC/MS/MS analysis, shinorine and porphyra-334 were detected from *Aph. flos-aquae* strains, but only shinorine was found in *Aph. gracile* strains.

Because of the lack of the standards, it was difficult to qualify the specific MAA concentration in the *Aphanizomenon* species, so the *M. aeruginosa* PCC7806 was used as a standard preparation source because of its high content of MAAs. However, the results showed that both shinorine and porphyra-334 were also detected in the bloom-forming *M. aeruginosa* strain culture, and the purity of these two MAAs was relatively high, which was a finding that contradicted previous articles that stated that there was only shinorine in the *M. aeruginosa* PCC7806 culture [25]. However, this result is coincident with Hu’s later findings of two MAAs from the *Microctsis* sp.T30 in Lake Erie [26].

The MAAs showed a widespread and mosaic distribution among those bloom-forming cyanobacteria because of their genus classification and living habitat differentiation. In most cases, cyanobacteria accumulate MAAs when they are exposed to harsh environments [3,47,56,57], and different MAA-producing cyanobacteria in various habitats [5] have different functions. The MAAs extracted from a marine green alga Chlamydomonas hedleyi (shinorine, porphyra-334 and mycosporine-Gly) showed potential in relation to anti-skin aging activity in human keratinocyte cells [58]. In addition, shinorine and porphyra-334 could competitively inhibit the activity of collagenase [59] and promote wound healing [60]. The two MAAs could also serve as a novel UV sunscreen to protect and stimulate the growth of human fibroblast cells [60,61]. Moreover, the UV absorption extracts from the red algae porphyra yezoensis (mainly shinorine and porphyra-334) could also block the photodimer formation [62] showed an anti-photoaging activity, and eliminated the free radical activity; thus, it has more benefits to the pharmaceutical industries [12]. Additionally, MAAs have already been globally commercialized in the skin-care and cosmetic industries, such as in the Helioguard 365, M-rose, and Fikia [12,63]. We can transform useless and discarded algal biomass into beneficial products, which represents a helpful recycling direction for cyanobacterial utilization.

The MAAs were shown to act as a UV sunscreen, but it is still unknown whether the MAAs really play a protective role for the *Aphanizomenon* strains against UV radiation, which will be elucidated in future studies.

## 5. Conclusions

This study revealed the classification origin of the four isolated *Aphanizomenon* strains. The results showed that there were two kinds of MAAs discovered in the cultured *Aphanizomenon* strains, shinorine and porphyra-334, but only shinorine existed in *Aph. gracile* strains. *M. aeruginosa* PCC7806 was used as an MAA standard source, and the purity of shinorine and porphy-334 could be 90.2% and 85.4%, respectively. As for the final MAA concentration, the *Aph. flos-aquae* shinorine and porphyra-334 concentrations were 0.307–0.385 µg/mg and 0.111–0.136 µg/mg, respectively, with the highest total concentration being 0.443–0.497 µg/mg dry weight. The *Aph. gracile* strains synthesized only shinorine with a concentration of 0.003–0.049 µg/mg, and the productivity of the *Aph. flos-aquae* MAAs was higher than that of *Aph. gracile*.

## Figures and Tables

**Figure 1 molecules-27-01734-f001:**
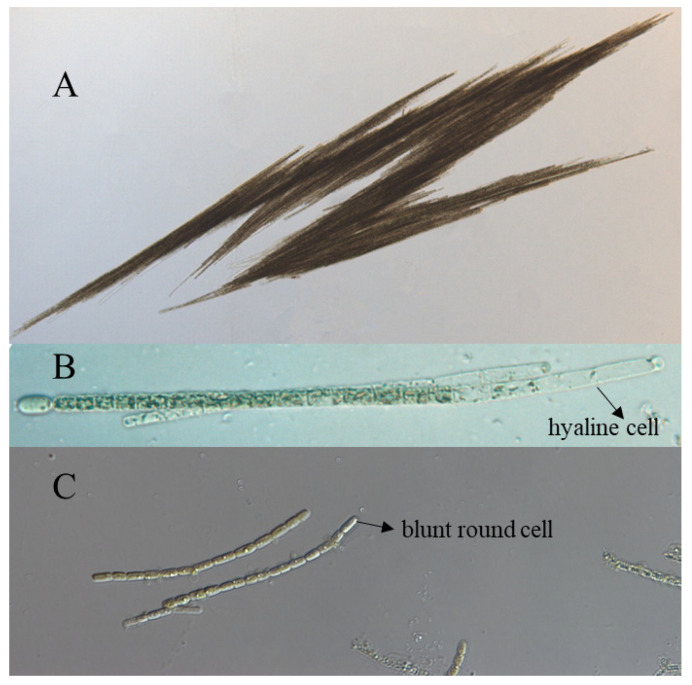
Morphological characteristics of the two different filamentous cyanobacteria. (**A**) outside sample of *Aphanizomeon flos-aquae*; (**B**) *Aph. flos-aquae* had fascicle-like assemblages with elongated and hyaline terminal cells. (**C**) *Aph. gracile*, blunt round terminal cells characters.

**Figure 2 molecules-27-01734-f002:**
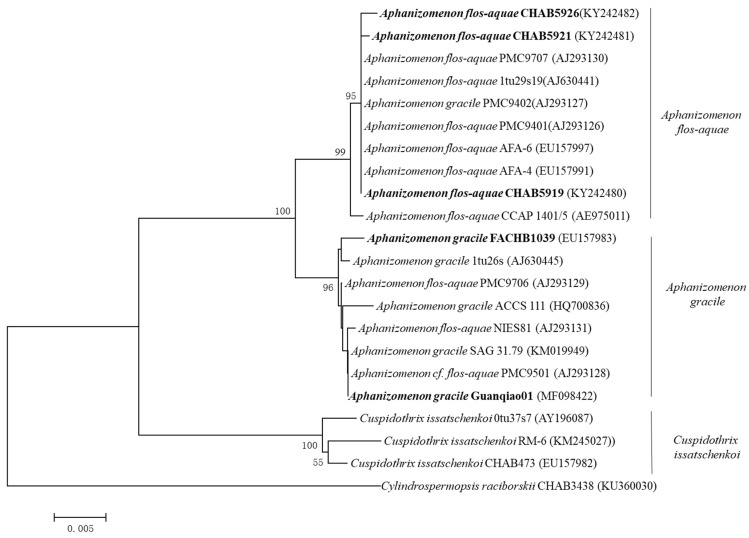
Phylogenetic analysis of the 16S rRNA gene sequences of *Aphanizomenon* strains used in this study. GenBank accession numbers were presented in the parenthesis.

**Figure 3 molecules-27-01734-f003:**
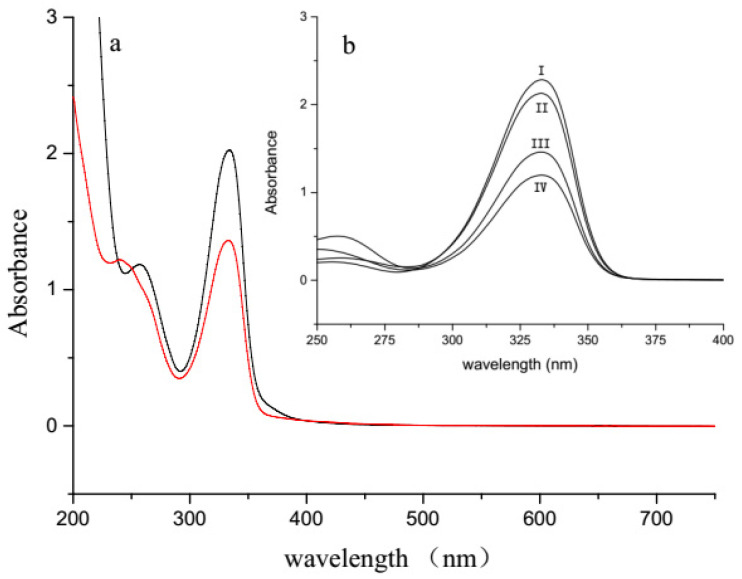
Full-wavelength scanning of the extraction of *M. aeruginosa* PCC7806 and *Aph. flos-aquae* CHAB5921 dry alga powder. (**a**) UV-Vis absorption spectrum of the methanolic extraction of PCC7806 (black line) and CHAB5921 (red line) strains. Both have the same absorption at 334 nm. (**b**) Collecting the two fractions of the strain PCC7806 and CHAB5921. Both showed the same absorption at 334 nm. Line I: 7806-Shinorine; Line II: 5921-Shinorine; Line III: 5921-Porphyra-334; Line IV: 7806-Porphyra-334.

**Figure 4 molecules-27-01734-f004:**
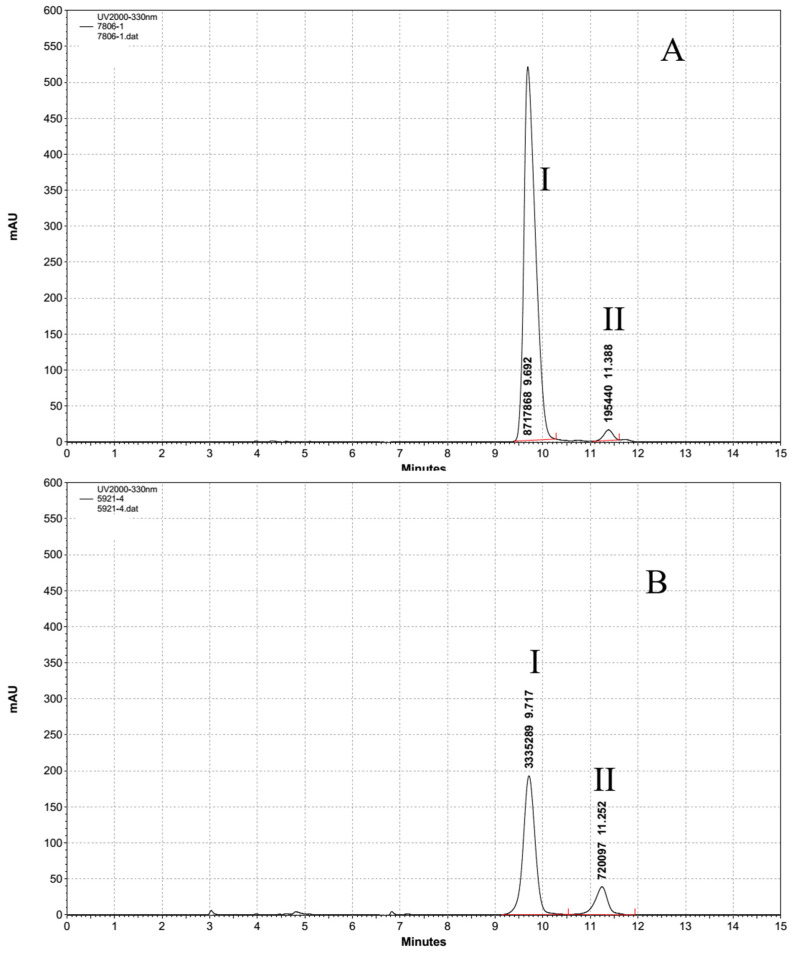
HPLC chromatography analysis of the purified MAAs of two strains. Both *M aeruginosa* PCC7806 (**A**) and *Aph. flos-aquae* CHAB5921 (**B**) extraction showed the same two characteristic peaks at 9.7 min and 11.3 min. I: shinorine content, II: porphyra-334 content.

**Figure 5 molecules-27-01734-f005:**
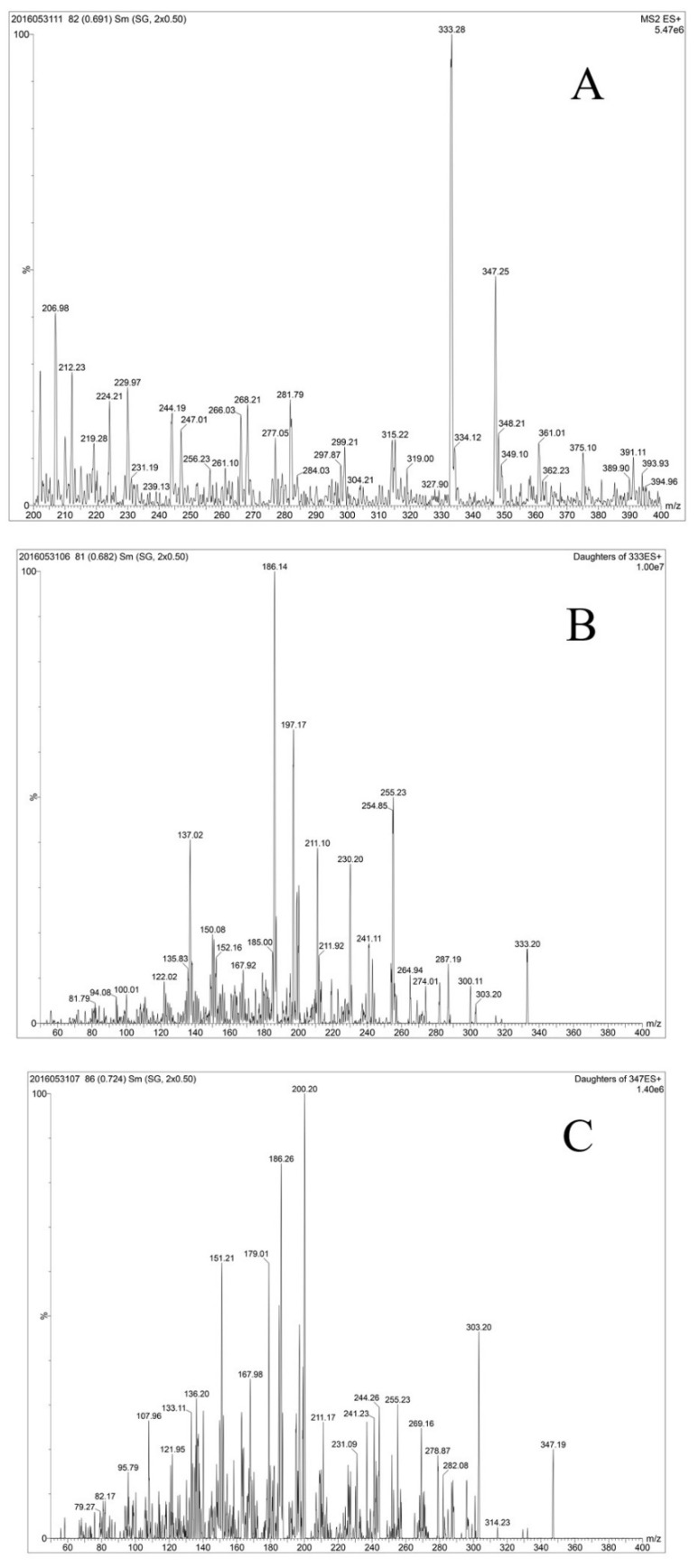
UPLC-MS-MS analysis of the MAAs in *Aph. flos-aquae* CHAB5921. (**A**) MS1 of the extraction of strains CHAB5921. (**B**) MS2 daughter fragment analysis of the fragment 333-shinorine. (**C**) MS2 daughter fragment analysis of fragment 347 porphyra-334.

**Table 1 molecules-27-01734-t001:** The GenBank accession numbers of *Aphanizomenon* strains in this study.

Species	Stains	Origin	Accession Number (16S rRNA)
*Aphanizomenon gracile*	Guanqiao01	Guanqiao fish pond, China	MF098422
	FACHB1039	Lake Dianchi, China	EU157983
*Aphanizomnon flos-aquae*	CHAB5919	Lake Dianchi, China	KY242480
	CHAB5921		KY242481
	CHAB5926		KY242482

Strain Guanqiao01 was from the Guanqiao fishpond in Wuhan, China. The other four strains FACHB1039, CHAB5919, 5921, 5926 were isolated from Lake Dianchi in Kunming city, China.

**Table 2 molecules-27-01734-t002:** Fragments of shinorine and porphyra-334 in different organisms.

Substance	[M + H]^+^	Fragment Patterns	Organism	Reference
shinorine	333.00	152,168,185,186,197,211,230,236,241,255, 256,274,287,300,318	standards	[19]
		137,168,185,186,197,211.230,241,255	*Nodularia spumigena*	[45]
		137,150,168,185,186,197,211,230,241,255, 274,287,300	*Aphanizomenon* *flos-aquae*	This study
porphyra-334	347.00	137,152,168,186,188,200,210,230,244,255,270,283,303,314	standards	[19]
		137,151,168,185,186,197,200,243,303	*Nodularia spumigena*	[45]
		137,151,168,179,186,200,244,255,269,283, 303	*Aphanizomenon* *flos-aquae*	This study

**Table 3 molecules-27-01734-t003:** MAAs concentration in *Aphanizomenon* strains (the MAAs content of all strains is expressed as µg /mg, ND means not detected).

Species	Strains	Shinorine (µg/mg)	Porphyra-334 (µg/mg)	Total MAAs (µg/mg)
*Aphanizomenon* *flos-aquae*	CHAB5919	0.307 ± 0.016	0.136 ± 0.005	0.443 ± 0.02
	CHAB5921	0.385 ± 0.005	0.112 ± 0.001	0.497 ± 0.005
	CHAB5926	0.378 ± 0.003	0.111 ± 0.002	0.489 ± 0.004
*Aphanizomenon* *gracile*	FACHB1039	0.049 ± 0.001	ND	0.049 ± 0.001
	Guanqiao01	0.003 ± 0.001	ND	0.003 ± 0.001

## Data Availability

The data presented in this study are available in this article.

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
