# Peer review of "Occurrence of Mycosporine-like Amino Acids (MAAs) from the Bloom-Forming Cyanobacteria Aphanizomenon Strains"

_molecules, 2022, doi:10.3390/molecules27051734_

Round 1

Reviewer 1 Report

The manuscipt titled "Occurrence of mycosporine-like amino acids (MAAs) from the bloom-forming cyanobacteria Aphanizomenon strains" is adherent to the aims of the Moleules Journal.  

The introduction provide a sufficient background and include all relevant references; the research design is appropriate; the methods are well described. The results are clearly presented and the conclusions are supported by the results. The reference section needs some integration/corrections. See below.

ref. 11 "Biotechno" add "l";

refs 2, 5, 9, 10: "J" add dot;

ref 18 and 22: "Annu" and "Arch" add dot, respectively;

ref. 32: "methods" change Methods;

ref. 34: "Microbiol", add dot;

ref. 35: "Bacterio" add "l."

refs 46, 49, 50, 62: "Photoche" "Photobio" add "m" and "l" ,respectively;

Table 2: To correct "fagment ..." to "fragment"

Author Response

Response to Reviewer 1 Comments

Point 1: The manuscipt titled "Occurrence of mycosporine-like amino acids (MAAs) from the bloom-forming cyanobacteria Aphanizomenon strains" is adherent to the aims of the Moleules Journal.The introduction provide a sufficient background and include all relevant references; the research design is appropriate; the methods are well described. The results are clearly presented and the conclusions are supported by the results.

Response 1: First of all, we want to thank the reviewer for reviewing our manuscript.These comments and suggestions help to improve the quality of this study. Thanks for your encouragement, we will keep trying.

Point 2: The reference section needs some integration/corrections. See below.

Response 2: Thanks for the reviewer’ carefully examination,we will check the all the references once again according to the comments with the assistance of both artificial work and specific software instruction.

Point 3: ref. 11 "Biotechno" add "l";

Response 3: ref.11 “Biotechno” was replaced by “Biotechnol.”. Revised Details seeing the new version of manuscript.

Point 4: refs 2, 5, 9, 10: "J" add dot;

Response 4: we have added dot to “J.” in ref 2,5,9,10.

Point 5: ref 18 and 22: "Annu" and "Arch" add dot, respectively;

Response 5: A dot was added to “Annu.” and “Arch.”in ref 18 and 22.

Point 6: ref. 32: "methods" change Methods;

Response 6: we have repalced the ref.32 “methods” to “Methods”

Point 7: ref. 34: "Microbiol", add dot;

Response 7: we have added a dot to the ref.34 “Microbiol.”.

Point 8: "Bacterio" add "l."

Response 8: ref. 35 “Bacterio” was replaced by “Bacteriol.”.

Point 9: refs 46, 49, 50, 62: "Photoche" "Photobio" add "m" and "l" ,respectively;

Response 9: "Photoche" and "Photobio" were replaced by “Photochem.” and “Photobiol.” in refs 46,49,50 ,62,respectively.

Point 10: Table 2: To correct "fagment ..." to "fragment"

Response 10: we will correct the “fagment…”to “fragment”, thanks again for the reviewer’ effort.

Reviewer 2 Report

In this work, five species of Aphanizomenon were isolated and identified and then the production of MAAs by these strains was also evaluated.                                                                                         

Although it is an interesting topic, the manuscript has many flaws in the presentation of the data and it needs an extensive editing of English language.

The Materials and Methods need to be improved and additional information should be added because the description of the experimental conditions is not sufficient to reproduce the experiments. Some examples: concentrations of the different components for the PCR reactions (e.g dNTPs, genomic DNA). The biomass dry weight used for the MAAs extractions, concentration of the samples run on HPLC, phylogenetic analysis is poor.

I have some problems with the identification of the strains. The taxonomy of cyanobacteria is not an easy task and the authors do not display in the manuscript the enough data to support the identification at species level. The morphological description is poor and not compared with close species. Concerning the molecular data, the sequences were compared against NCBI database? How were selected the sequences for the phylogenetic analysis? Why the Microcystis was used as outgroup? Aphanizomenon are heterocystous strains, other outgroup could be used instead of Microcystis. The type species for both strains were used (gracile and flos-aquae)? How the authors defined these clusters? This information should be clear.

The authors showed some differences in the peaks (HPLC analyses). The same biomass dry weight was used for all strains and Microcystis? The differences observed could have be related with the efficiency of the extraction?

I encourage the authors to rewrite and improve the manuscript and resubmit it again.

Author Response

Response to Reviewer 2 Comments

Point 1: In this work, five species of Aphanizomenon were isolated and identified and then the production of MAAs by these strains was also evaluated. Although it is an interesting topic, the manuscript has many flaws in the presentation of the data and it needs an extensive editing of English language.

Response 1: Thanks for the evaluation and encouragement, the revised manuscript was sent to a professional English Service to have the manuscript re-edited for English language. Please find the attached certificate.

Point 2: The Materials and Methods need to be improved and additional information should be added because the description of the experimental conditions is not sufficient to reproduce the experiments. Some examples: concentrations of the different components for the PCR reactions (e.g dNTPs, genomic DNA). The biomass dry weight used for the MAAs extractions, concentration of the samples run on HPLC, phylogenetic analysis is poor.

Response 2: We are sorry for those points. We have added some descriptions of experimental conditions as suggested by the reviewer. We added some details contents in the phylogenetic analysis. Thanks for your favourable suggestions.

Point 3: I have some problems with the identification of the strains. The taxonomy of cyanobacteria is not an easy task and the authors do not display in the manuscript the enough data to support the identification at species level. The morphological description is poor and not compared with close species.

Response 3: Thanks for the point. We agree that the cyanobacterial taxonomy is not an easy task. We have added some morphological characters in the revised manuscript to ensure the taxonomic identity for the strains used in this study.

Point 4: Concerning the molecular data, the sequences were compared against NCBI database? How were selected the sequences for the phylogenetic analysis? Why the Microcystis was used as outgroup? Aphanizomenon are heterocystous strains, other outgroup could be used instead of Microcystis.

Response 4: Yes, we selected the NCBI strains with correct taxonomic belongings since many strains were not correctly identified. We changed the outgroup as Cylindrospermopsis in the phylogenetic tree to replace Microcystis since the genus Cylindrospermopsis is a filamentous and heterocytous cyanobacterial group . Thanks for the suggestion.

Point 5: The type species for both strains were used (gracile and flos-aquae)? How the authors defined these clusters? This information should be clear.

Response 5: We used the strains of type species in Aphanizomenon, but types strain in cyanobacteria is not mandatory since this is following the Nomenclature of Algae, Fungi and Plants, rather than the Nomenclature of Bacteria.

Point 6 : The authors showed some differences in the peaks (HPLC analyses). The same biomass dry weight was used for all strains and Microcystis? The differences observed could have be related with the efficiency of the extraction?

Response 6: We agreed with the viewer opinion . We used the other extractants to test their extraction efficiency, and found the methanol was better than other solvents such as aceton or ethanol. In this article, the MAAs shinorine and porphyra-334 were both discovered in both Microcystis and Aphanizomenon species, but the MAAs concentration in Microcystis was higher than Aphanizomenon with the same biomass dry weight, the different extraction efficiency may be related with the strains morphological character, which need a further investigation. Thanks for the reviewer’ comments.

Point 7: I encourage the authors to rewrite and improve the manuscript and resubmit it again.

Response 7: Thank you very much for your help and encouragement.

Reviewer 3 Report

Review for the manuscript-1595719  Zhang H.; Jiang,Y.G.; Zhou C.; Chen,Y.C.; Yu,G.L,; Zheng L.P.; Guan,H.L.; Li, R.H. Occurrence of mycosporine-like amino acids (MAAs) from the bloom-forming cyanobacteria Aphanizomenon strains. Molecules 2022, 27, x The results of the manuscript are well-discussed, methodologically correct and relevant. The manuscript complies with the rules of the journal "Molecules" and can be published with some minor changes. The authors complemented the existing information of the presence and composition of mycosporin-like amino acids in 5 strains of two species of cyanoprokaryotes of the genus Aphanizomenon. The specificity of species that caused 'blooming' events in different types of water bodies in China and its pharmacopoeial significance for practical use were discussed. However, it would be desirable to mention another achievement of the authors - molecular-taxonomic in the text of the manuscript (Fig. 2 – p. 199): The taxonomic revision of 18 strains of the Aphanizomenon genus (collection and natural material) according to molecular-biological criteria was conducted. Also the species affiliation was specified. According to the results, Aph. gracile strain AJ293127 belongs to the Aph. flos-aquae clade, and four strains of Aph. flos-aquae (AJ293128, PMC9501 (?), EU157983, AJ293131) - to the Aph. gracile clade.  

Author Response

Response to Reviewer 3 Comments

Point 1: The results of the manuscript are well-discussed, methodologically correct and relevant. The manuscript complies with the rules of the journal "Molecules" and can be published with some minor changes. The authors complemented the existing information of the presence and composition of mycosporin-like amino acids in 5 strains of two species of cyanoprokaryotes of the genus Aphanizomenon. The specificity of species that caused 'blooming' eventsin different types of water bodies in China and its pharmacopoeial significance for practical use were discussed.

Response 1: First of all, we want to thank the reviewer for reviewing our manuscript.The following comments and suggestions will help in improving the quality of current study. Thanks for your encouragement, we will keep working on it.

Point 2: However, it would be desirable to mention another achievement of the authors-molecular-taxonomic in the text of the manuscript(Fig.2–p.199): The taxonomic revision of 18 strains of the Aphanizomenon genus(collection and natural material) according to molecular-biological criteria was conducted. Also the species affiliation was specified. According to the results, Aph. gracile strain AJ293127 belongs to the Aph. flos-aquae clade, and four strains of Aph.flos-aquae (AJ293128, PMC9501 (?), EU157983, AJ293131) - to the Aph. gracile clade.

Response 2: Thanks for the reviewer’ carefully examination, we checked the all the strains names and their accession numbers. By the comparison with the 16S rRNA nucleic sequences in Genbank, we adjusted the name Aph. flos-aquae CHAB1039 to Aph. graile FACHB1039 (thses finds were also our laboratary mentioned work, seeing Ref.33), and the Aph. flos-aquae PMC9706 16S rRNA accession number was replaced byAJ293129.

The classification of Aphanizomenon species is not an easy task. Before the molecular-biological methods bringing into the classification system, mainstream named the algae was according to their morphological character, and the Aphanizomenon strains main feature changed when the culture varied, the typical fasicle-like assemblages in Aph. flos-aquae strains may become a single filament, then it is difficult to name the algae species, the Aph.flos-aquae PMC9706,PMC9501 and NIES81 were also belong to these situation, because the previous articles had named these strains to Aph.flos-aquae clade, for the purpose of respecting the predecessor’ work, afterwards articles did not change their original name, but the molecular-biological verified that Aph.flos-aquae PMC9706,PMC9501 and NIES81 were classified into the Aph.gracle clade. The new phylogenetic analysis of 16S rRNA gene sequences of main Aphanizomenon strains displayed below.
